analytical chemistry/nanotechnology

polypropylene glycol, cephradine, PPG-AuNPs, colorimetric sensor, environmental and biological samples

**Authors for correspondence:**
Muhammad Iqbal Bhanger
e-mail: dbhanger2000@gmail.com
Muhammad Imran Malik
e-mail: mimran.malik@iccs.edu

†Daim Asif Raja and Fazeelah Munir have equal contribution in this study and both should be considered as first authors.

This article has been edited by the Royal Society of Chemistry, including the commissioning, peer review process and editorial aspects up to the point of acceptance.

# Colorimetric sensing of cephradine through polypropylene glycol functionalized gold nanoparticles

Daim Asif Raja[1,†], Fazeelah Munir[1,†], Muhammad Raza Shah[1], Muhammad Iqbal Bhanger[1] and Muhammad Imran Malik[1,2]

[1]H.E.J. Research Institute of Chemistry, and [2]Third World Center for Science and Technology, International Center for Chemical and Biological Sciences (ICCBS), University of Karachi, Karachi 75270, Pakistan

MIM, 0000-0001-6942-0407

The development of metal nanoparticle-based facile colorimetric assays for drugs and insecticides is an emerging area of current scientific research. In the present work, polypropylene glycol was used for stabilization of gold nanoparticles (AuNPs) in a simple one-pot two-phase process and subsequently employed it for the specific detection of cephradine (CPH). The characterization of the prepared PPG-AuNPs was conducted through various analytical techniques such as UV-visible spectrophotometry, Fourier transform infrared spectroscopy, atomic force microscopy (AFM), zeta potential and zetasizer techniques. As the major target of the study, the stabilized PPG-AuNPs were employed for colorimetric detection of CPH and other drugs. Typical wine-red colour of PPG-AuNPs disappeared immediately and surface plasmon resonance band quenched by addition of CPH in the presence of several other interferents (drugs and salts) and in real samples. PPG-AuNPs permitted efficient, selective, reliable and rapid determination in a concentration range of 0.01–120 mM with a detection limit (LoD) of 11.0 mM. The developed sensor has the potential to be used for fast scanning of pharmaceutical formulations for quantification of CPH at production facilities.

# 1. Introduction

Antibiotics are penetrating into our ecosystem from various sources, continuously affecting marine and terrestrial environment as well as human metabolic activities [1]. These antibiotics and their subsequent metabolites are excreted from the human body after performing their intended functions [2]. Cephradine, CPH, ($C_{16}H_{19}N_3O_4S$) is one of the most commonly used antibiotics for respiratory and urinary tract infections [3]. It is a first-generation cephalosporin antibiotic that is specifically used for various bacterial infections relevant to skin, soft tissues and ear [4]. The presence of CPH in biological, pharmaceutical and environmental samples is inevitable owing to its widespread use worldwide. There are numerous analytical methods that can be employed for the determination of CPH in different environmental and pharmaceutical samples. These techniques include spectrophotometry [5], spectrofluorimetry [6], electrochemistry [7], HPLC [4] and electrophoresis [8]. However, these instrumental techniques require expensive instrumentation, well-established laboratory set-up, tedious sample pre-treatment steps, long analysis time and trained operators. Hence, there is a room for a facile protocol for CPH detection without involving high-tech instrumentation.

With the emergence of nanotechnology, plenty of research approaches are investigated and numerous applications of nano-size particles are explored [9–14]. Metal nanoparticles (MNPs) have been widely used for diverse applications owing to their unique optical and catalytic properties compared to their bulk counterparts. These metal nanoparticles have been used in drug delivery formulations, antimicrobial agents, cosmetics, environmental applications, biological sensing and catalysis [15,16]. Particularly, gold nanoparticles (AuNPs) have been widely used in sensing, biomedical and bioanalytical applications owing to the peculiar correlation of their size and colour [17].

Several different methods have been reported for the formation of AuNPs such as Turkevich method, Brust-Schiffrin, electrochemical method and seeding growth method [18]. In this context, the stabilization of AuNPs with some ligand is an important aspect of their selectivity and sensitivity towards a particular analyte [19–22]. These stabilizing agents not only prevent the aggregation of the NPs but also induce different functionalities at the surface of MNPs that can be exploited for further specialized applications.

The stabilized AuNPs have exciting surface plasmonic properties and found applications in various research fields such as sensors [23], drug delivery [24], biomedical imaging [25], cancer treatment [26] and in diagnosis and therapy of several other diseases [27]. AuNPs have been employed for the colorimetric detection of biological and chemical compounds due to their large surface area and specific distance-dependent optical properties [12,22,28–33].

An especially important optical property of stabilized AuNPs is their specific response to the addition of certain analytes in them. This specific response may be the result of electrostatic attraction, Van der Waal interactions and/or induction forces. The incorporation of stabilizing agents, having different functional groups, on the AuNPs may induce specificity and selectivity with regard to any analyte owing to the presence of complementary functional groups on the analyte. The addition of such an analyte may result in visual change in the colour of AuNPs solution that may be further evaluated by the shift in their surface plasmon resonance (SPR) band [11,34,35]. Another important phenomenon could be an aggregation of nanoparticles by the addition of external species which would result in the disappearance of typical wine-red colour of AuNPs and the disappearance of the SPR band [36,37]. In this context, our group have reported numerous chemo-sensors based on metal nanoparticles by using different stabilizing agents for their specific and selective response to different analytes such as insecticides, drugs, metal ions, etc. [22,29,36,38,39].

Among numerous stabilizing agents for NPs, polyethylene glycol (PEG) has been extensively used due to its high biocompatibility, excellent water solubility and non-biofouling characteristics [40]. Polypropylene glycol (PPG) is structurally close to PEG with an exception of a pendant methyl group. The extra methyl group decreases the hydrophilicity of the PPG and induces hydrophobic character [41]. PPG is a stable, non-toxic, biodegradable and biocompatible polymer that is used for the preparation of many biomaterials and other biomedical products [42]. The presence of an ether group in the chain makes it a suitable candidate for the stabilization of metal nanoparticles [43]. Herein, we propose the use PPG as stabilizing agent for AuNPs and to explore its specificity and selectivity for different analytes in the context of sensors.

In this study, we present a rapid, easy, efficient and one-step synthesis method of PPG stabilized AuNPs (PPG-AuNPs) by using tetrachloroauric (III) acid trihydrate ($HAuCl_4 \cdot 3H_2O$) as a gold source, and sodium borohydride ($NaBH_4$) as a reducing agent. Characterization of freshly prepared PPG-AuNPs was conducted by multiple analytical techniques such as atomic force microscopy (AFM) for

surface morphology, zeta sizer for size distributions and zeta potential for overall charge on the particles before and after addition of an analyte. Fourier transform infrared (FT-IR) and UV-visible spectroscopy were used to evaluate the presence of functional groups and their subsequent interaction with the analyte. The synthesized PPG-AuNPs were employed as a specific colorimetric assay for CPH in environmental (water), pharmaceutical (drugs) and biological (plasma, urine and serum) samples. The potential applications of the proposed colorimetric assay for CPH for environmental, biological and pharmaceutical samples are demonstrated in the presence of naturally occurring interfering species.

# 2. Experimental section

## 2.1. Materials and instruments

PPG ($Mn = 4000 \text{ g mol}^{-1}$) and $NaBH_4$ were purchased from Sigma-Aldrich, Germany, $HAuCl_4 \cdot 3H_2O$ from Merck Chemicals, Germany, and HPLC grade methanol from Tedia, USA. CPH standard was taken from local pharmaceutical company and commercial samples of the drug were purchased from a local pharmacy.

The glassware was washed with 10% nitric acid to minimize the contamination risk, and afterwards, rinsed with distilled water followed by drying in the oven. The pH meter from Laqua Horbia (pH 1300) was used having glass working and Ag/AgCl reference electrode. A double-beam spectrophotometer (CECIL 7400) was used to record UV-visible spectra in the region of 300 to 800 nm by using a quartz cuvette having a path length of 1 cm. FT-IR (Bruker Vector 22) having deuterated triglycine sulfate detector was used to record the spectra in the region of $400–4000 \text{ cm}^{-1}$ using KBr disc and 10 scans with a spectral resolution of $0.1 \text{ cm}^{-1}$ were recorded.

To determine the particle size and zeta potential, nano-ZSP (Malvern Instruments) (zeta sizer) was used. AFM (Agilent 5500) in tapping mode was used to record topographical images of PPG-AuNPs. For the sample preparation, one drop of the sample was placed on a silicon wafer that is air-dried for 24 h. The triangular nitride silicon cantilever (Veeco, model MLCT-AUHW) was used for the analysis of the sample under a spring constant value of $0.1 \text{ N m}^{-1}$.

## 2.2. Preparation of PPG-AuNPs

Solutions of $HAuCl_4 \cdot 3H_2O$ (0.25 mM) and $NaBH_4$ (5.0 mM) were prepared in deionized water and the solution of PPG-4000 (0.1 mM) was prepared in methanol. The ratio of the prepared solutions for preparation of PPG-AuNPs was kept $1:15:0.1$ (PPG: $HAuCl_4 \cdot 3H_2O$: $NaBH_4$). PPG solution was stirred continuously while adding $HAuCl_4 \cdot 3H_2O$ solution to the flask followed by extra stirring for an hour. Afterwards, the solution of $NaBH_4$ was added dropwise and the mixture was stirred for another 20 min. The appearance of the wine-red colour indicates the formation of PPG-AuNPs. The effect of different external parameters which include temperature, pH and the presence of electrolyte on the stability PPG-AuNPs was evaluated.

## 2.3. Application of PPG-AuNPs as a colorimetric sensor

The aqueous solutions of CPH and other drugs having 0.1 mM concentration were prepared and mixed with the equal volume of the solution of PPG-AuNPs to evaluate its effect on the visual change in colour followed by evaluation of variation in the SPR band by UV-visible spectroscopy.

### 2.3.1. Analysis in tap water

Tap water was taken from the University of Karachi. The above procedure was followed except for the preparation of CPH solution which was prepared in tap water. The solution of PPG-AuNPs and CPH were mixed, and the change in the UV-Vis spectrum was observed.

### 2.3.2. Analysis in human blood plasma and urine

A blood sample was taken from the healthy human by using a venous puncture procedure with his consent at the Center for Bioequivalence Studies and Clinical Research (CBSCR), International Center for Chemical and Biological Sciences (ICCBS). Plasma was extracted from the blood by centrifuging it at 4000 r.p.m. for 5 min at 25°C.

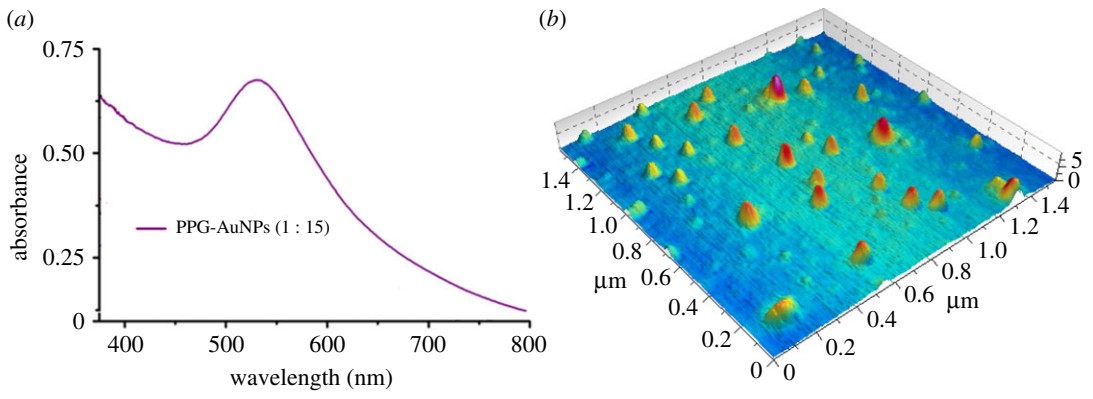

**Figure 1.** (*a*) UV-visible spectra of PPG-AuNPs at the optimized ratio of PPG and HAuCl$_4$ . 3H$_2$O (1 : 15); (*b*) three-dimensional AFM image of PPG-AuNPs.

For the analysis, one control and other experimental solution of plasma were prepared. The control contains 2.0 ml of plasma, 4.0 ml of PPG-AuNPs, and filled up to 10.0 ml with distilled water. In the experimental solution, 0.1 mM CPH was added, and afterwards, UV-visible spectra of both solutions were recorded. The same procedure was followed for the urine sample.

## 2.4. Analysis in pharmaceutical drug

For the determination of CPH content in the pharmaceutical formulation, commercial drug samples of Velosef (250 and 500 mg) of Glaxo Smith Kline were purchased from the local market and a calculated amount was added into 2.0 ml of PPG-AuNPs. After 5 min, the solutions were analysed by UV-visible spectroscopy. A similar process was employed to other commercial samples and concentrations were recorded using single-point analysis. In parallel, the commercial samples were also analysed by HPLC-UV and UV-Vis spectroscopy for absorbance at 254 nm.

# 3. Results and discussion

## 3.1. Synthesis and characterization of PPG-AuNPs

PPG is a polyether with a pendant methyl group in the repeat unit. The ether groups in the polymer chain induce dipole and polarity in the polymer which makes it a potential candidate as a stabilizing agent for metal nanoparticles. A typical SPR band of AuNPs exists in the absorption range of 500–550 nm. The stoichiometric equivalence of the groups responsible for stabilization on the stabilizing agent and AuNPs is important for optimum stabilization. In this case, a ratio of 1 : 15 for solutions of PPG and HAuCl$_4$ · 3H$_2$O (v/v) was found to be optimum, figure 1*a*. The absorption decreased while going away from this ratio in either direction [22]. The formation of AuNPs at the optimized ratios is further confirmed by AFM imaging, figure 1*b*. The average size and zeta potential value of these prepared AuNPs were calculated using dynamic light scattering (DLS), and were found to be 104.6 nm and −8.0 mV respectively, figure 5*a*.

The next important aspect to evaluate any prepared AuNPs is their stability when exposed to different experimental variables. Temperature treatment at 100°C increased the absorption, which is an indication of enhanced stability after thermal treatment, figure 2*a*. Furthermore, PPG-AuNPs remain stable for more than a month at 4°C. Another aspect to be evaluated is the stability of NPs in the presence of different electrolytes. Sodium chloride in a concentration range of 0.01 to 5 M in PPG-AuNPs resulted in a decrease in the intensity of SPR band, which may be attributed to aggregation of the metal ions in the presence of free chloride (Cl$^{-1}$) ions, figure 2*b* [22]. The absorption band is not affected to a large extent while using NaCl concentration below 0.5 M. NaCl concentrations above this resulted in a significant drop in the intensity of the absorption band. On the same lines, the stability of PPG-AuNPs is evaluated as a function of the pH. The maximum intensity of the SPR band was found at pH 6. Changing pH in either direction leads to a decrease in the SPR band intensity; however, pH 5 of NPs is selected for further studies as it is the original pH of prepared NPs, figure 2*c* [22]. Hence, PPG-AuNPs are found to be stable at different possible external parameters during their applications.

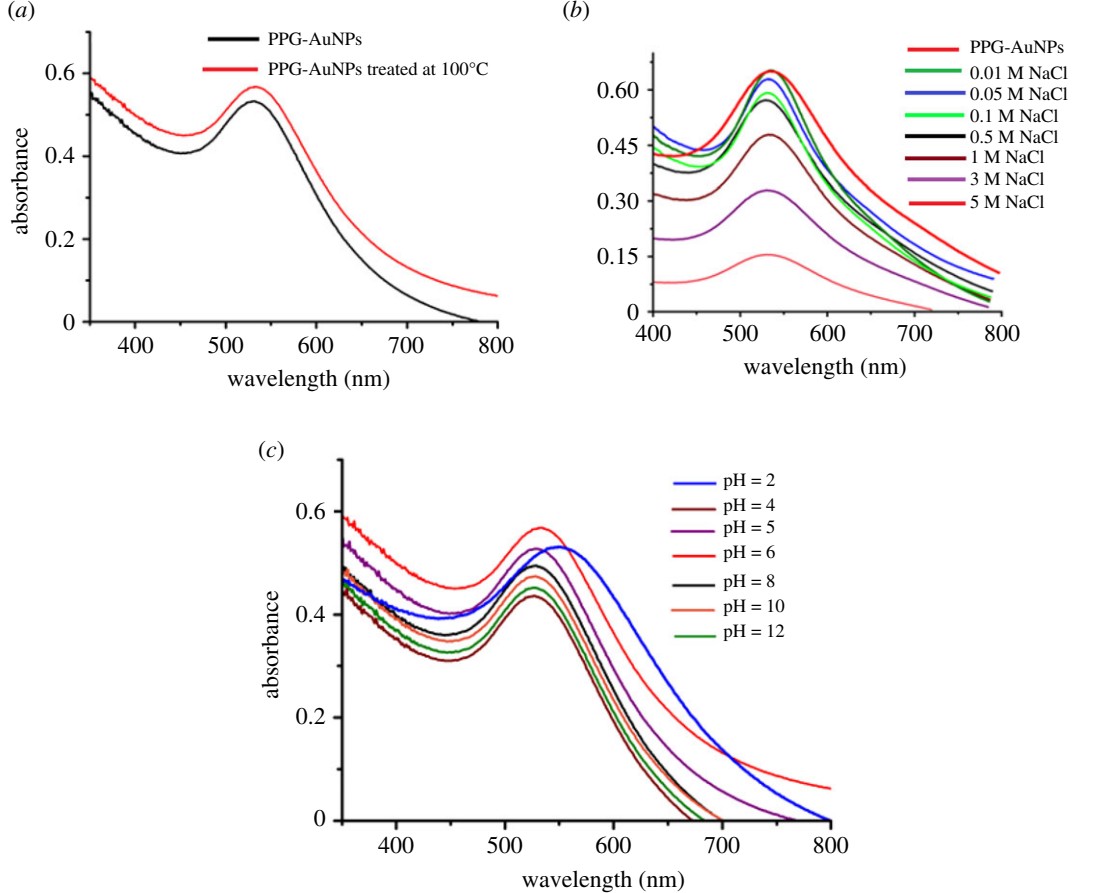

**Figure 2.** Stability of PPG-AuNPs through UV-visible spectroscopy (*a*) before and after incubation at 100°C; (*b*) after adding different concentrations of NaCl; (*c*) at different pH. Some parts of figure reproduced with permission of Elsevier from [22].

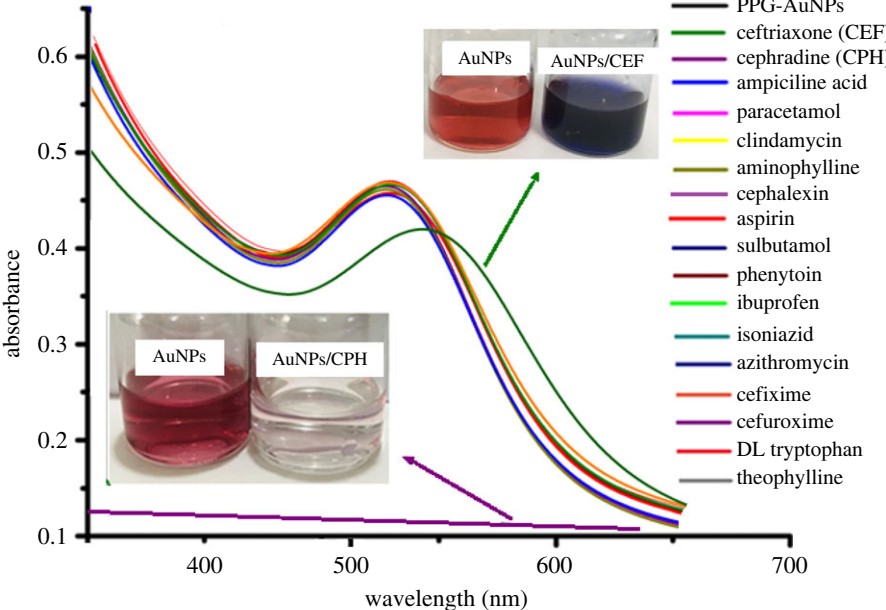

**Figure 3.** UV-visible spectra of PPG-AuNPs before and after addition of different drugs; inset show the colour change after addition of CEF and CPH. A part of figure is reproduced from [22] with permission of Elsevier.

## 3.2. PPG-AuNPs and drug interaction

Interaction of drugs of different nature with PPG-AuNPs was evaluated by mixing equal volume of the solution of the drug and the PPG-AuNPs. The addition of 15 tested drugs named in figures 3 and 4 did

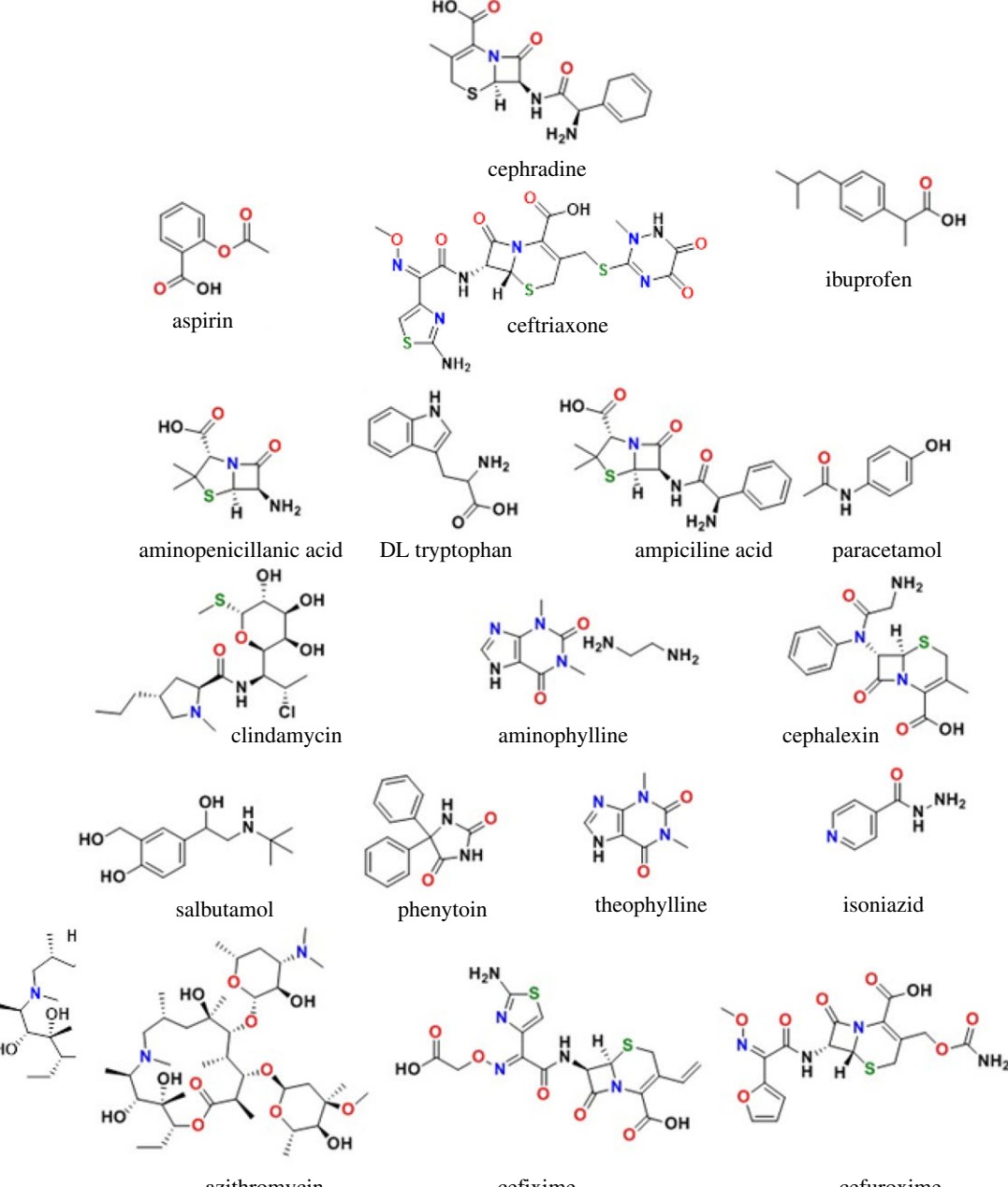

**Figure 4.** Structure of CPH and other tested drugs in this study.

not bring any physical change in colour and the UV absorption band of PPG-AuNPs. However, the addition of ceftriaxone resulted in a colour change from wine-red to dark blue, and the SPR band shifted from 532 to 572 nm [22]. Furthermore, the addition of CPH resulted in an abrupt disappearance of colour and quenching of the SPR band of PPG-AuNPs, figure 3. The abrupt disappearance of colour and quenching of the SPR band is attributed to the aggregation of AuNPs. Structures of the drugs used in this study are shown in figure 4.

Moreover, AFM, zeta potential and zeta sizer were used to evaluate the changes in PPG-AuNPs after the addition of CPH. A clear increase in size that is leading to aggregation of PPG-AuNPs is visible after the addition of CPH, figure 5a-I, and figure 5b-I. The average size of PPG-AuNPs was found to be 104.6 nm with a polydispersity index (PDI) of 0.395 that increased to 180.1 nm with a PDI of 0.201 after the addition of CPH, figure 5a-II, and figure 5b-II. Another important aspect is the net charge on the NPs that keeps them away from each other to avoid their aggregation. Net charge away from zero in either direction induces stability to the NPs. The net potential on the surface of PPG-AuNPs was found to be −8.0 mV which is an indication of reasonable stability of PPG-AuNPs. The net surface

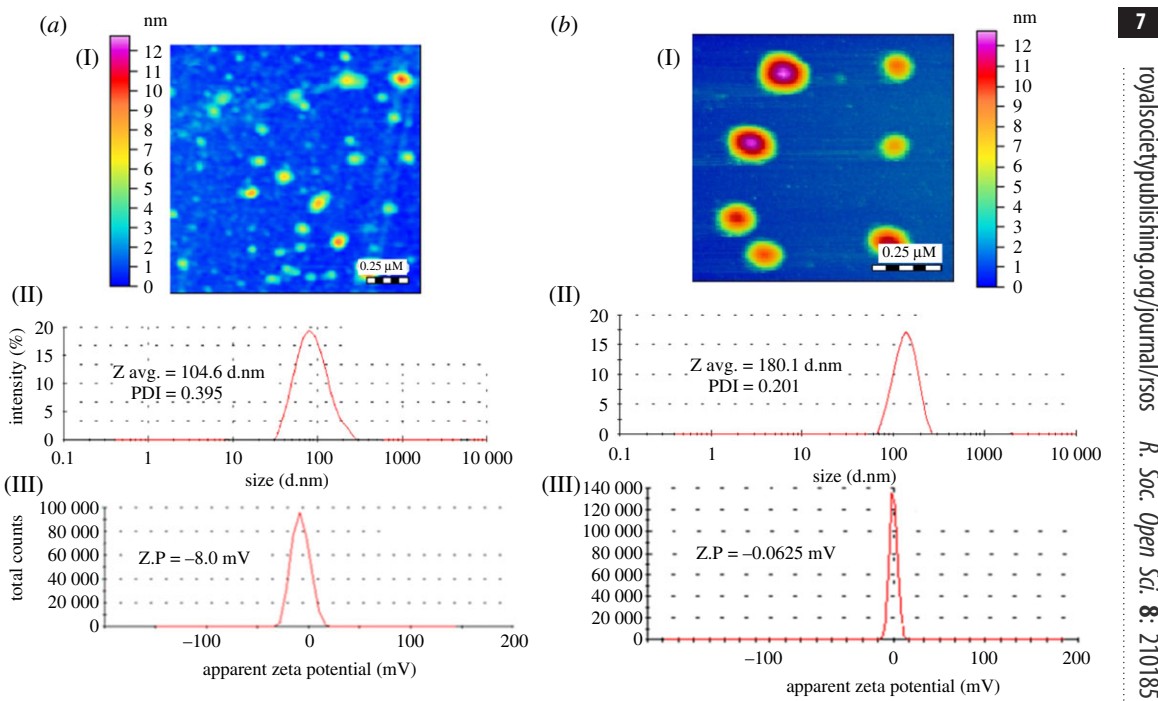

**Figure 5.** Comparison of PPG-AuNPs before and after addition of CPH by (I) AFMs, (II) average size, (III) the zeta potential of (*a*) PPG-AuNPs; (*b*) PPG-AuNPs/CPH; d.nm (diameter in nanometre); PPG (0.1 mM): HAuCl$_4$ · 3H$_2$O (0.25 mM): NaBH$_4$ (5.0 mM) = 1 : 15 : 0.1 (v/v); pH ∼ 5.

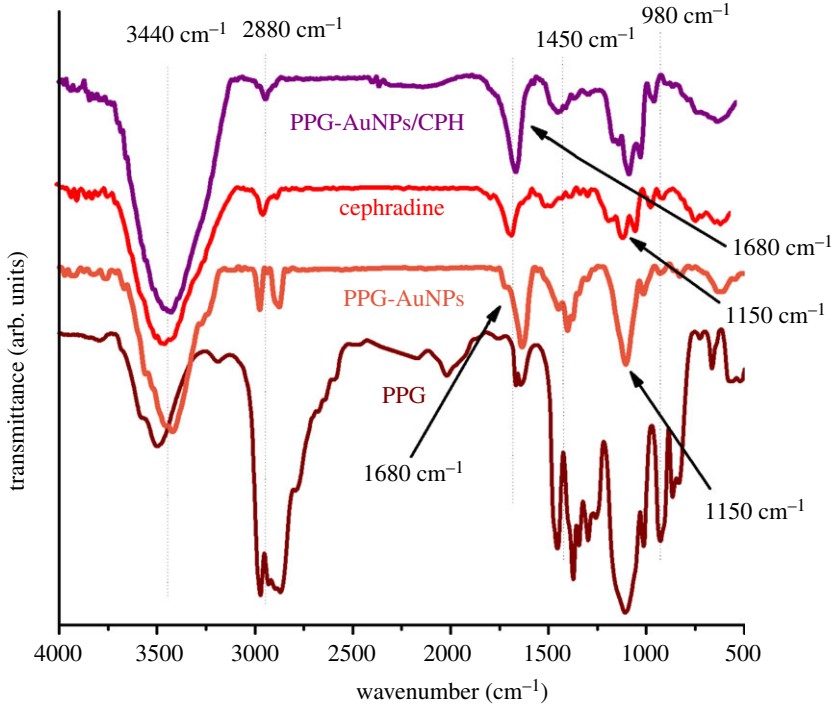

**Figure 6.** FTIR spectra of PPG, PPG-AuNPs, CPH and PPG-AuNPs/CPH.

charge approaches zero after the addition of CPH which strongly suggests the aggregation of PPG-AuNPs, figure 5*a*-III, and figure 5*b*-III.

FTIR analysis of PPG, PPG-AuNPs, CPH and PPG-AuNPs/CPH was carried out to determine the forces responsible for the stabilization of AuNPs and the interaction of the drug with PPG-AuNPs.

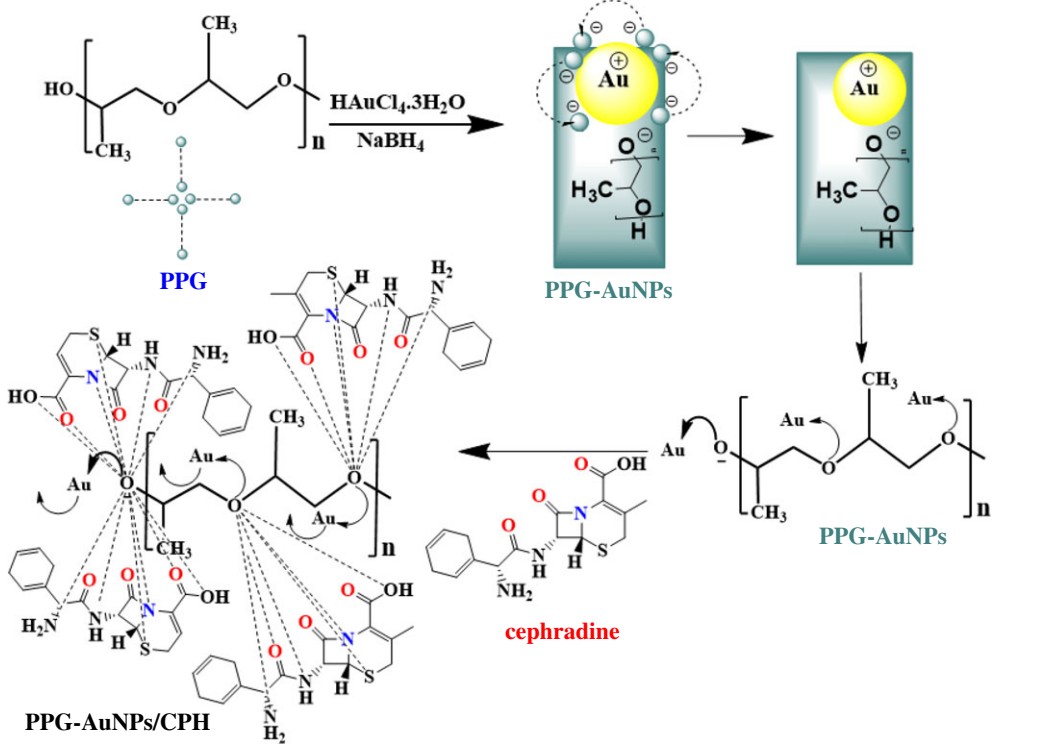

**Scheme 1.** Schematic representation of AuNPs formation through steric stabilization by the PPG and drug recognition (CPH) of PPG-AuNPs/CPH.

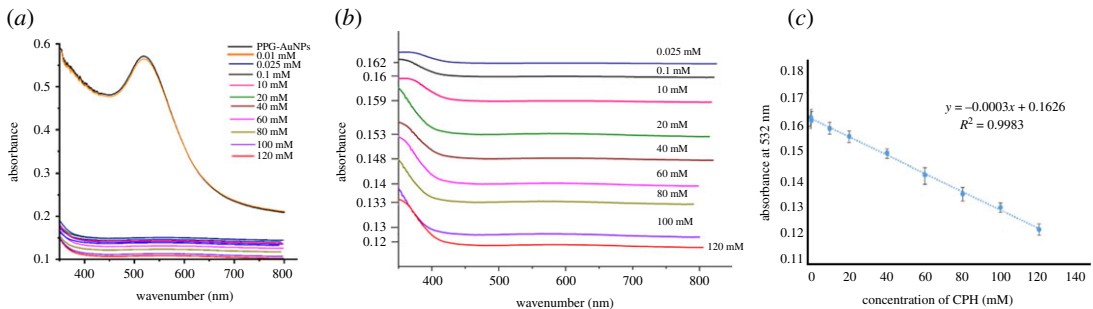

**Figure 7.** (*a*) UV-Vis spectra of PPG-AuNPs after addition of the variable amount of CPH; (*b*) magnified version of figure 7*a* in a range of 0.12 to 0.162 absorbance; (*c*) absorbance intensity as a function of the concentration of CPH.

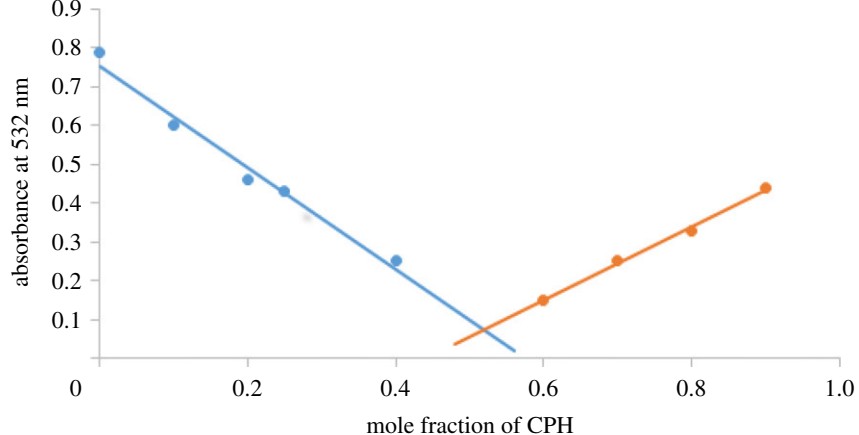

**Figure 8.** Job plot for the binding ratio of PPG-AuNPs and CPH.

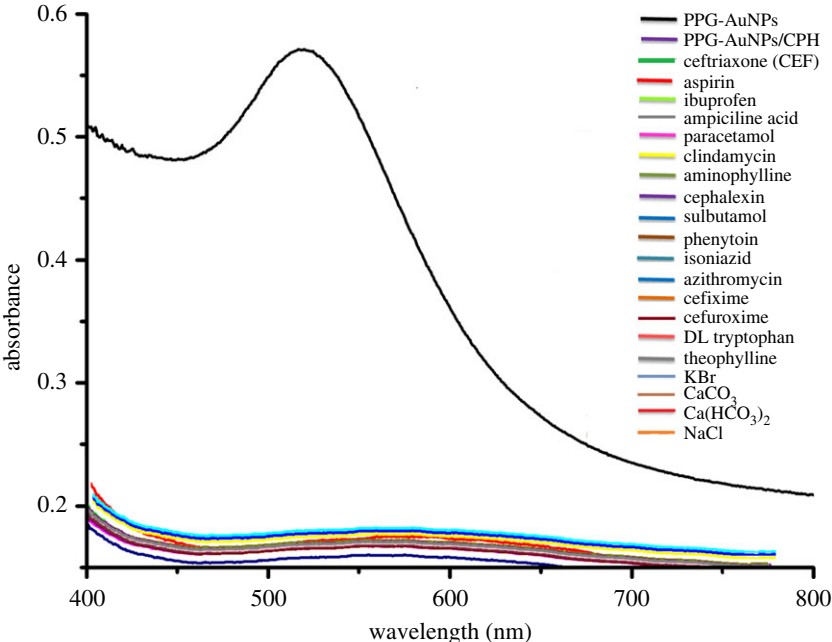

**Figure 9.** UV-Vis spectra showing the effect of interfering drugs and salts on CPH detection by PPG-AuNPs.

The peaks of PPG at 3440 cm$^{-1}$ (attributed to hydroxyl group stretching) shifted to 3340 cm$^{-1}$ in the PPG-AuNPs spectrum which indicates the capping of AuNPs [22]. The appearance of a peak at 1680 cm$^{-1}$ confirms the formation of Au-O bond. The addition of CPH in PPG-AuNPs did not affect the representative IR peak of AuNPs at 1680 cm$^{-1}$ which indicates the absence of any direct interaction of CPH with AuNPs. Nonetheless, the intensity of the peak at 1150 cm$^{-1}$ for –C–O decreases after the addition of CPH which strongly indicates the interaction of CPH with oxygen present in the PPG. Meanwhile, in spectra of PPG-AuNPs and PPG-AuNPs/CPH, a clear change in the peak behaviour was observed in a range of 2800 to 3000 cm$^{-1}$, which indicates the change in stretching of CH$_3$ and CH$_2$, figure 6.

The addition of CPH in the PPG-AuNPs resulted in an immediate disappearance of the typical wine-red colour and quenching of the SPR band. As shown above, electrostatic interactions are responsible for the stabilization of AuNPs by PPG. The addition of CPH in stabilized PPG-AuNPs induces other competitive forces between ether groups of PPG and functional groups present on CPH. CPH contains functional groups having the ability to donate electrons such as (C=O, C–O, N–H), which may interact strongly with the oxygen in the PPG chain leading to aggregation of AuNPs, scheme 1. Another reason for this abrupt quenching may be the transfer of electrons from AuNPs to cyclohexa-1, 4-diene moiety in CPH which is not present in other tested drugs.

## 3.3. Quantitative recognition of CPH through PPG-AuNP-based sensor

By recording UV-Vis spectra of PPG-AuNPs/CPH at different concentrations of CPH, the analytical capacity of the developed method is evaluated, figure 7a. All of the used concentrations resulted in quenching of the SPR band. A careful analysis of the quenched peaks after the addition of CPH revealed a direct correlation between drug concentration and the extent of quenching, figure 7b. The absorbance at 532 nm is inversely proportional to the amount of CPH added. As a next step, a calibration curve is constructed by plotting absorbance at 532 nm as a function of the concentration of CPH, figure 7c. A reasonable linear correlation is found in the concentration range of 0.025–120 mM with the regression constant ($R^2$) value of 0.9983. The addition of 0.01 mM CPH solution did not affect the SPR band of PPG-AuNPs. The limit of detection [calculated by formula 3.0 × (s.d. of intercept/slope)] of the proposed sensor was found to be 11.0 mM [44,45].

Immediate disappearance of the typical wine-red colour from PPG-AuNPs by the addition of CPH indicated the presence of CPH, which is further confirmed by the disappearance/quenching of SPR band. The coupling stoichiometry between PPG-AuNPs and CPH as obtained by Job plot is found to be 1 : 1, figure 8.

**Table 1.** Comparison of the analytical methods for the detection of CPH.

| method/materials | linear range | LoD[a] | LoD (mM)[b] | sample | remarks | ref. |
|---|---|---|---|---|---|---|
| zero-crossing derivative spectrophotometry | 2.0–56.0 µg ml$^{-1}$ | 0.16 µg ml$^{-1}$ | 0.00045 | saline physiological serum and physiological serum-containing glucose | conditions optimization, no interference study, complex statistical calculations required, no study in biological samples | [13] |
| atomic absorption spectrometric determination | 5–70 µg ml$^{-1}$ | 6.69 µg ml$^{-1}$ | 0.01914 | drug samples | lengthy conditions optimization, cost intensive, required heavy instrumentation, no interference study, no study in environmental and biological samples, trained operators required | [46] |
| electrochemical study of the degradation product | 10$^{-7}$–10$^{-6}$ mol l$^{-1}$ | 0.5 × 10$^{-7}$ mol l$^{-1}$ | 0.00005 | CPH solution | instrument-based method, selective electrodes required, limited accuracy, no interference study, trained operators required, no study in environmental and biological samples | [7] |
| fluorescent supramolecular tweezers | 0.1–5 µM | 1 µM | 0.001 | mixture of drugs | complicated protocol, proper laboratory setup required, expensive instruments required, condition optimization required, no study in environmental and biological samples | [47] |
| capillary zone electrophoresis | 93.8–6255.6 mg ml$^{-1}$ | 5.0 µg ml$^{-1}$ | 0.01431 | mixture of drugs | tedious sample preparation, costly internal standards, long analysis time, optimization issues, proper laboratory setup and trained operators required | [8] |
| spectrofluorimetric method | 0.1–5.0 µg ml$^{-1}$ | 1.09 × 10$^{-2}$ ± 3.64 × 10$^{-3}$ µg ml$^{-1}$ | 0.000028 | commercial formulations | optimization of conditions, sophisticated instrumentation, trained operators required, cost intensive method | [6] |
| fluorosurfactant-capped gold nanoparticles | 2.0–10.0 mg ml$^{-1}$ | 0.8 µg ml$^{-1}$ | 0.00228 | pharmaceutical formulations | no study in environmental and biological samples, usage of costly reagents, no interference studies in presence of similar nature drugs and electrolytes | [48] |

(*Continued.*)

**Table 1.** (*Continued.*)

| method/materials | linear range | LoD[a] | LoD (mM)[b] | sample | remarks | ref. |
|---|---|---|---|---|---|---|
| high-performance liquid chromatographic method | 0.2 to 30.0 μg ml$^{-1}$ | 0.2 μg ml$^{-1}$ | 0.000572 | human plasma samples | exclusive instrumentation, trained operators, long analysis time, lengthy sample preparation, optimization issues, long analysis time, expensive standards required | [4] |
| spectrophotometric/PPG-AuNPs method | 0.025–120 mM | 11.0 mM | 11.0 | pharmaceutical formulations, water, blood plasma, urine, serum | simple procedure, reasonable sensitivity, high selectivity, short analysis time, basic instrumentation, economical, on-spot visual indication, no pre-treatment of the sample | this method |

[a]LoD in the units as given in the reference.

[b]LoD unit converted to mM.

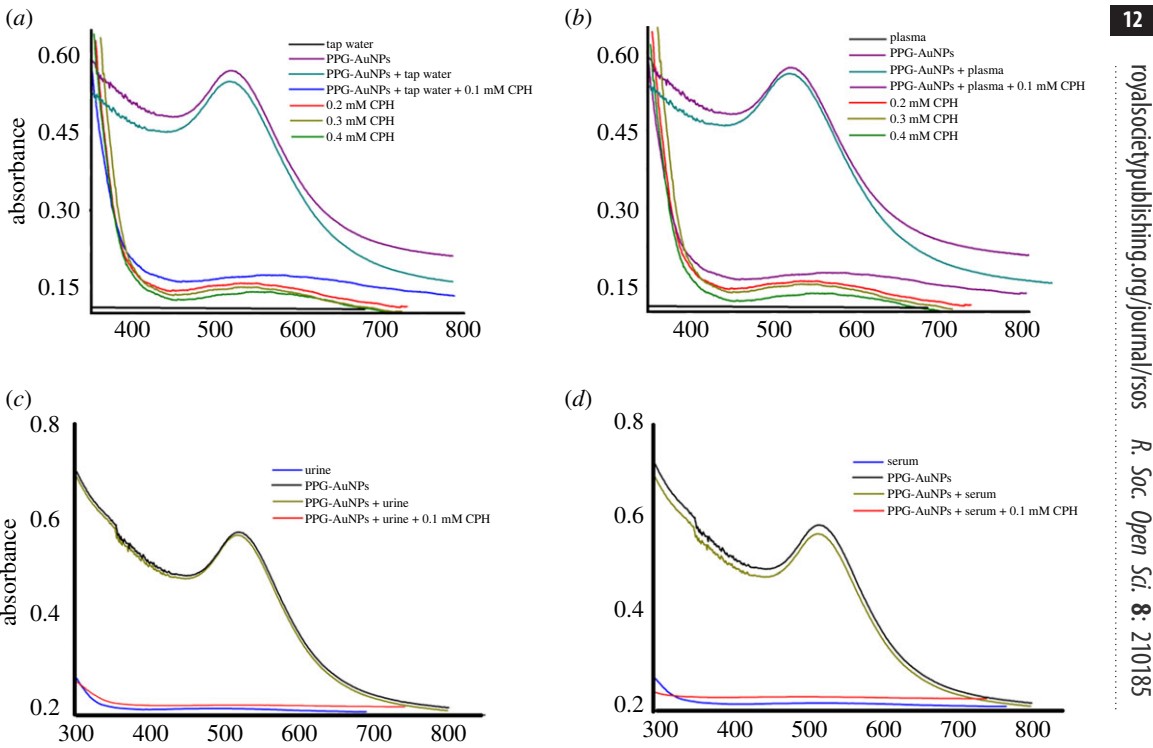

**Figure 10.** Effect of addition of CPH on absorption band of PPG-AuNPs (*a*) tap water; (*b*) blood plasma; (*c*) urine; (*d*) serum.

**Table 2.** Quantitative analysis of CPH containing commercial samples with regard to claimed content of CPH.

| sample | mass of drug (mg) | amount of CPH as obtained by different techniques (mg) | | | manufacturer claim CPH (mg) |
|---|---|---|---|---|---|
| | | UV-Vis at 254 nm | HPLC-UV | PPG-AuNPs | |
| Velosef (500 mg), (GSK) | 590 ± 1 | 490 ± 2 | 495 ± 3 | 430 ± 2 | 500 |
| Velosef (250 mg), (GSK) | 295 ± 2 | 235 ± 1 | 240 ± 1 | 215 ± 1 | 250 |

The efficiency of any developed method must be validated in the presence of other interfering drugs and salts in the context of real sample analysis. The developed sensor for CPH in this study was found to be robust in the presence of several other interfering drugs and salts. The presence of other drugs namely aspirin, ibuprofen, ampiciline acid, paracetamol, clindamycin, aminophylline, cephalexin, salbutamol, phenytoin, isoniazid, azithromycin, cefixime, cefuroxime, DL tryptophan, theophylline and salts namely potassium bromide, calcium carbonate, calcium bicarbonate and sodium chloride having concentration equal to CPH did not have any pronounced effect on the efficiency of the developed colorimetric CPH sensor, figure 9. Hence, the efficiency of the proposed quenching sensor for CPH was not affected by the presence of other competing drugs and salts which is an advantage in context of real sample analysis.

CPH analysis has been reported using many instrumental techniques such as HPLC, ELISA, voltammetry, NMR and GC-MS in different environmental and biological samples, table 1. Although the limit of detection of the instrumental methods is lower compared to the colorimetric sensor proposed in this study, these methods require sophisticated instrumentation, highly equipped laboratory set-up, expensive reagents and trained operators. On the contrary, the proposed method in this study is facile, fast, without any requirement of sample preparation and produces on-spot results by visual demonstration. Furthermore, the obtained limit of detection is in a fairly low range in the context of practical applications. The unaffected efficiency of the proposed CPH assay in presence of different interferents (drugs and salts) is an added advantage for such colorimetric sensors in the context of real sample analysis in minimum time without any high-tech instrumentation.

All the above studies were carried out in DI water. Moreover, the authenticity of interference study and validity of PPG-AuNP-based proposed assay for CPH in context of real samples was evaluated by employing the optimized protocols for tap water, urine, serum and blood plasma samples. Immediate disappearance of wine-red colour of PPG-AuNPs and quenching of the typical UV-Vis peak confirms the applicability of the proposed CPH assay for all real samples in the presence of natural interferents, figure 10. The extent of quenching in the real samples follows the same calibration curve and hence the assay is proved to be reliable for the environmental and biological samples in tap water, urine, serum and blood plasma.

As the next step, the developed CPH assay was used for quantification and quality control of commercial drugs containing CPH. An appropriate amount of the commercial drug as per claims of the manufacturer was added to the PPG-AuNPs. The immediate disappearance of the wine-red colour of the PPG-AuNPs gave the first indication of the presence of CPH. The next step is the evaluation of the quantitation capability of the proposed colorimetric sensor. The extent of quenching in the UV absorption band, evaluated by the absorbance after addition of the drug, is compared with the calibration curve, figure 7c. The amount of CPH as obtained by the calibration curve is fairly close to the manufacturer claim and analysis of the same sample by independent HPLC-UV and UV-Vis spectroscopy, table 2.

The proposed colorimetric assay for CPH in this study is robust and applicable to samples containing different interfering species and for quick quality control in the production facilities. The advantages of the proposed colorimetric assay for CPH include fast, facile, on-spot analysis of drug contamination in environmental and biological samples without any involvement of high-tech instrumentation.

# 4. Conclusion

The uniquely developed colorimetric sensing based on PPG-AuNPs has proved to be a rapid and sophisticated one-pot quantitative assay for CPH. Characterization of PPG-AuNPs, CPH and PPG-AuNPs/CPH was done through zeta sizer, AFM, FT-IR and UV-visible spectroscopy. The synthesized PPG-AuNPs were found to be resistant to various external variables such as temperature, pH and the presence of electrolytes. Typical wine-red colour of AuNPs immediately disappears by the addition of CPH and SPR band of AuNPs quenched. The proposed method has a linear dynamic range from 0.025 to 120 mM with a limit of detection of 11.0 mM. The presence of other drugs and ions did not have any pronounced effect on the sensing ability of the PPG-AuNP-based assay for CPH. The proposed method can be used for environmental, biological and pharmaceutical samples. Reasonable selectivity, the simple procedure of preparation and application, a relevant dynamic range and limit of detection demonstrate the potential of PPG-AuNP-based CPH detection method for practical applications without any involvement of high-tech instrumentation. Furthermore, the proposed method can be employed for a quick screening of pharmaceutical formulations in the context of quantification of CPH on production facilities.

Ethics. The blood sample was collected after ethical approval from the institute committee of Center for Bioequivalence Studies and Clinical Research (CBSCR), International Center for Chemical and Biological Sciences (ICCBS).

Data accessibility. Data are available at https://doi.org/10.5061/dryad.6wwpzgmxt.

Authors' contributions. D.A.R. and F.M. performed experiments and wrote the initial version of the manuscript. D.A.R. and F.M. have equal contribution in this study and both should be considered as first authors. M.R.S. provided facilities for the performed work along with intellectual input; M.I.B. provided intellectual input and help writing the manuscript. M.I.M. conceived the idea and finalized manuscript

Competing interests. The authors declare no conflict of interest.

Funding. There is no external funding to report for this project.

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
