## [Peer Review File · Royal Society Open Science]

Review History

RSOS-210185.R0 (Original submission)

Review form: Reviewer 1

Is the manuscript scientifically sound in its present form?

No

Are the interpretations and conclusions justified by the results?

No

Is the language acceptable?

No

Do you have any ethical concerns with this paper?

No

Have you any concerns about statistical analyses in this paper?

No

Recommendation?

Major revision is needed (please make suggestions in comments)

Comments to the Author(s)

The paper can be published only after very major revision reflecting comments inserted as yellow notes into attached pdf (see Appendix A) of submitted manuscript. English must be improved.

Review form: Reviewer 2**Is the manuscript scientifically sound in its present form?**

Yes

Are the interpretations and conclusions justified by the results?

Yes

Is the language acceptable?

Yes

Do you have any ethical concerns with this paper?

No

Have you any concerns about statistical analyses in this paper?

No

Recommendation?

Accept with minor revision (please list in comments)

Comments to the Author(s)

In this work, the authors described an optical probe for cephradine detection based on polypropylene glycol functionalized gold nanoclusters. The manuscript is logical and the data are sound, which makes this work informative for the readership of Royal Society Open Science. However, there still exist some queries need to be considered.

1. In Fig.5(III), please specify the solution condition (concentration, pH, etc.) due to zeta potential is sensitive to the solution condition.
2. In Fig.7C and Fig. 8, for more clearly indicating the ordinate, the name of ordinate should be "absorbance at 532 nm" instead of just "absorbance".
3. In order to prove the accuracy of proposed method, please include the recovery and relative standard deviation (RSD) in Table 2.
4. As the authors said the PPG increase the hydrophobic character compared to PEG, the PPG-AuNPs may have the lower stability in aqueous solution, what about the stability of PPG-AuNPs solution?

Decision letter (RSOS-210185.R0)

Dear Dr Malik:

Title: Colorimetric sensing of Cephadrine through Polypropylene glycol functionalized Gold Nanoparticles
Manuscript ID: RSOS-210185

The editor assigned to your manuscript has now received comments from reviewers. We would like you to revise your paper in accordance with the referee and Subject Editor suggestions which can be found below (not including confidential reports to the Editor). Please note this decision does not guarantee eventual acceptance.

Please submit your revised paper before 04-Apr-2021. Please note that the revision deadline will expire at 00.00am on this date. If we do not hear from you within this time then it will be assumed that the paper has been withdrawn. In exceptional circumstances, extensions may be possible if agreed with the Editorial Office in advance. We do not allow multiple rounds of revision so we urge you to make every effort to fully address all of the comments at this stage. If deemed necessary by the Editors, your manuscript will be sent back to one or more of the original reviewers for assessment. If the original reviewers are not available we may invite new reviewers.

RSC Associate Editor:
Comments to the Author:
(There are no comments.)

RSC Subject Editor:
Comments to the Author:
(There are no comments.)

Reviewers' Comments to Author:
Reviewer: 1

Comments to the Author(s)
The paper can be published only after very major revision reflecting comments inserted as yellow notes into attached pdf of submitted manuscript. English must be improved.

Reviewer: 2

Comments to the Author(s)
In this work, the authors described an optical probe for cephadrine detection based on polypropylene glycol functionalized gold nanoclusters. The manuscript is logical and the data are sound, which makes this work informative for the readership of Royal Society Open Science. However, there still exist some queries need to be considered.

1. In Fig.5(III), please specify the solution condition (concentration, pH, etc.) due to zeta potential is sensitive to the solution condition.
2. In Fig.7C and Fig. 8, for more clearly indicating the ordinate, the name of ordinate should be "absorbance at 532 nm" instead of just "absorbance".
3. In order to prove the accuracy of proposed method, please include the recovery and relative standard deviation (RSD) in Table 2.
4. As the authors said the PPG increase the hydrophobic character compared to PEG, the PPG-AuNPs may have the lower stability in aqueous solution, what about the stability of PPG-AuNPs solution ?

Author's Response to Decision Letter for (RSOS-210185.R0)

See Appendix B.

Decision letter (RSOS-210185.R1)

Dear Dr Malik:

Title: Colorimetric Sensing of Cephadrine through Polypropylene Glycol Functionalized Gold Nanoparticles
Manuscript ID: RSOS-210185.R1

It is a pleasure to accept your manuscript in its current form for publication in Royal Society Open Science. The chemistry content of Royal Society Open Science is published in collaboration with the Royal Society of Chemistry.

RSC Associate Editor
Comments to the Author:
(There are no comments.)

Reviewer(s)' Comments to Author:

Appendix A**ROYAL SOCIETY
OPEN SCIENCE****Colorimetric sensing of Cephadrine through Polypropylene glycol functionalized Gold Nanoparticles**

Journal:	Royal Society Open Science
Manuscript ID	RSOS-210185
Article Type:	Research
Date Submitted by the Author:	12-Feb-2021
Complete List of Authors:	Raja, Daim Asif; University of Karachi International Center for Chemical and Biological Sciences Munir, Fazeelah; University of Karachi International Center for Chemical and Biological Sciences Shah, Muhammad Raza; University of Karachi International Center for Chemical and Biological Sciences Bhanger, Muhammad Iqbal; University of Karachi International Center for Chemical and Biological Sciences Malik, Muhammad Imran; University of Karachi International Center for Chemical and Biological Sciences
Subject:	Analytical chemistry < CHEMISTRY, Nanotechnology < CHEMISTRY
Keywords:	Polypropylene glycol, Cephadrine, PPG-AuNPs, colorimetric sensor, environmental and biological samples
Subject Category:	Chemistry

Author-supplied statements

Relevant information will appear here if provided.

Ethics

Does your article include research that required ethical approval or permits?:

This article does not present research with ethical considerations

Statement (if applicable):

CUST_IF_YES_ETHICS :No data available.

Data

It is a condition of publication that data, code and materials supporting your paper are made publicly available. Does your paper present new data?:

Yes

Statement (if applicable):

https://datadryad.org/stash/share/TF8Xb4NqiQ8CeD9PbSN2C0jq5zk_t5Qe9758r4SMj_A

Conflict of interest

I/We declare we have no competing interests

Statement (if applicable):

CUST_STATE_CONFLICT :No data available.

Authors' contributions

This paper has multiple authors and our individual contributions were as below

Statement (if applicable):

Daim Asif Raja performed experiments and wrote the initial version of the manuscript,
Fazeelah Munir performed experiments and wrote the initial version of the manuscript,
Muhammad Raza Shah provided facilities for the performed work along with intellectual input,
Muhammad Iqbal Bhangar provided intellectual input and help writing manuscript, Muhammad
Imran Malik conceived the idea and finalized manuscript

1
2
3
4
5
6
7
8
9
10
11
12
13
14
15

**Colorimetric sensing of Cephadrine through Polypropylene glycol functionalized
Gold Nanoparticles**

Daim Asif Raja, Fazeelah Munir, Muhammad Raza Shah, Muhammad Iqbal Bhangar*,
Muhammad Imran Malik**

H.E.J. Research Institute of Chemistry, International Centre for Chemical and Biological
Sciences (ICCBS), University of Karachi, Karachi 75270, Pakistan

E-mail: *dbhanger2000@gmail.com; **mimran.malik@iccs.edu

16
17

Abstract

18
19
20
21
22
23
24
25
26
27
28
29
30
31
32
33
34
35
36
37
38
39
40
41

Development of metal nanoparticle based facile colorimetric assays for drugs and insecticides is an emerging area of scientific research. In the present work, polypropylene glycol (PPG) was utilized for stabilization of AuNPs (Gold nanoparticles) by using simple one-pot two-phase process and subsequently employed for the specific detection of Cephadrine (CPH). The characterization of the prepared PPG-AuNPs was conducted through various analytical techniques such as UV-visible, Fourier Transform Infra-Red (FT-IR) Spectroscopy, Atomic Force Microscopy (AFM), zeta potential and zetasizer. As the major target of the study, the stabilized PPG-AuNPs were employed for colorimetric detection of CPH and other drugs. Typical wine red color of PPG-AuNPs disappeared immediately and SPR band quenched by addition of CPH in presence of several other interferents (drugs and salts) and in real samples. PPG-AuNPs permitted efficient, selective, quantitative and rapid recognition in concentration range of 0.01-120 mM with a detection limit of 10.98 mM. The developed sensor has potential to be used for quick scanning of pharmaceutical formulations for quantification of CPH at production facilities.

42
43
44
45

Keywords: Polypropylene glycol; Cephadrine; PPG-AuNPs; colorimetric sensor; environmental and biological samples

46
47
48
49

Author Contribution: Daim Asif Raja and Fazeelah Munir have equal contribution in this study and both should be considered as first authors.

1. Introduction

Antibiotics are penetrating into our ecosystem  various sources, ~~that is~~ continuously affecting marine and terrestrial environment as well as human metabolic activities.¹ These antibiotics and their subsequent metabolites are excreted from human body after performing their intended functions.² Cephadrine ($C_{16}H_{19}N_3O_4S$) is one of the most commonly used antibiotic for respiratory and urinary tract infections.³ It is a first generation cephalosporin antibiotic that is specifically used for various bacterial infections relevant to skin, soft tissues, and ear.⁴ The presence of cephradine (CPH) in biological, pharmaceutical, and environmental samples is inevitable owing to its wide spread use worldwide. There are numerous analytical methods that can be employed for the determination of CPH in different environmental and pharmaceutical samples. These techniques includes spectrophotometry,⁵ spectrofluorimetry,⁶ electrochemistry,⁷ HPLC,⁴ and electrophoresis.⁸ However, these instrumental techniques require expensive instrumentation, well-established lab set-up, tedious sample pretreatment steps, long analysis time, and trained operators. Hence, there is a room for a facile protocol for CPH detection without involving high-tech instrumentation.

With the emergence of nanotechnology, plenty of research approaches are investigated and numerous applications of nano-size particles are explored.⁹⁻¹⁴ Metal nanoparticles have been widely used due to their unique optical and catalytic properties compared to their bulk counterpart . These nanoparticles have been widely used in drug-delivery formulations, antimicrobial agents, cosmetics, environmental applications, biological sensing, and catalysis.^{15, 16} Particularly, AuNPs have been widely used in sensing, biomedical, and bioanalytical applications.¹⁷

Several different methods have been reported for the formation of AuNPs such as Turkevich method, Brust-Schiffrin, electrochemical method, and  seeding growth method.¹⁸ In this context, stabilization of AuNPs with some ligand is an important aspect for their selectivity and sensitivity towards a particular analyte.¹⁹⁻²² These stabilizing agents not only prevent the aggregation of the NPs but also induce different functionalities at the surface of MNPs that can be exploited for further specialized applications.

The stabilized AuNPs have exciting surface plasmonic properties and found applications in various research fields such as sensors,²³ drug delivery,²⁴ biomedical imaging,²⁵ cancer treatment,²⁶ and in diagnosis and therapy of diseases.²⁷ Gold nanoparticles (AuNPs) have been employed for the

1
2
3 colorimetric detection of biological and chemical compounds due to their large surface areas and
4 specific distance dependent optical properties.^{12, 22, 28-33}

5
6
7 An especially important optical property of stabilized AuNPs is their specific response to addition
8 of certain analytes ~~in them~~. This specific response may be result of electrostatic attraction, 9 van der
10 waal interactions, and/or induction forces. The incorporation of stabilizing agents having different
11 functional groups on the AuNPs may induce specificity and selectivity with regard to any analyte
12 owing to presence of complementary functional groups on the analyte. The addition of such an
13 analyte may result in visual change in the color of AuNPs solution that may  further ~~be~~ evaluated
14 by the shift in their SPR band.^{11, 34, 35} Another important phenomenon could be aggregation of
15 nanoparticles by addition of external species which would result in disappearance of typical red
16 wine color of AuNPs and disappearance of SPR band.^{36, 37} In this context, our group have reported
17 numerous ~~of~~ chemo sensors based on metal nanoparticles by using different stabilizing agents for
18 their specific and selective response  different analytes such as insecticides, drugs, metal ions
19 etc.^{22, 29, 36, 38, 39}

20
21
22 Among numerous capping agents for NPs, polyethylene glycol (PEG) has been extensively used
23 due to its high biocompatibility, excellent water solubility, and non-biofouling characteristics.⁴⁰
24 Polypropylene glycol (PPG) is structurally close to PEG with an exception of a pendant methyl
25 group. The extra methyl group decrease the hydrophilicity of the PPG and induce hydrophobic
26 character.⁴¹ PPG is a stable, non-toxic, biodegradable, and biocompatible polymer that is used for
27 the preparation of many biomaterials and other bio-medical products.⁴² The presence of an ether
28 group in the chain makes it a suitable candidate for stabilization  metal nanoparticles.⁴³ In this
29 context, we would like to use PPG as stabilizing agent for AuNPs and to explore its specificity and
30 selectivity for different analytes in  context of sensors.

31
32
33 In this study we present a rapid, easy, efficient and one-step synthesis method of PPG stabilized
34 AuNPs (PPG-AuNPs) ~~polymers~~ by using tetrachloroauric (III) acid trihydrate ($\text{HAuCl}_4 \cdot 3\text{H}_2\text{O}$) as
35 gold source and sodium borohydride (NaBH_4) as a reducing agent. Characterization of freshly
36 prepared PPG-AuNPs was conducted by multiple analytical techniques such as AFM for surface
37 morphology, Zeta sizer for size distributions, and Zeta potential for overall charge on the particles
38 before and after addition of  analyte. FTIR and UV-visible spectroscopy were used to evaluate the
39 presence of functional group and their subsequent interaction with  analyte. The synthesized PPG-

AuNPs were employed in a specific colorimetric assay for CPH in environmental (water), pharmaceutical (commercial) and biological (plasma, urine, and serum) samples. The potential applications of the proposed colorimetric assay for CPH for environmental, biological, and pharmaceutical samples demonstrated in the presence of naturally occurring interfering species.

2. Experimental Section

2.1. Materials and Instruments

Polypropylene glycol ($M_n = 4000$ g/mol) and NaBH_4 were purchased from Sigma-Aldrich, Germany, $\text{HAuCl}_4 \cdot 3\text{H}_2\text{O}$ from Merck Chemicals, Germany, and HPLC grade methanol from Tedia, USA. CPH standard was taken from local pharmaceutical company and commercial samples of the drug were purchased from a local pharmacy.

The glassware was washed with 10% nitric acid to minimize the contamination risk and afterward, rinsed with distilled water followed by drying in the oven. The pH meter from Laqua Horbia (pH 1300) was used having glassy working and Ag/AgCl as a reference electrode. A double beam spectrophotometer (CECIL 7400) was used to record UV-visible spectra in the vibration region of 300 to 800 nm by using a quartz cuvette having a path length of 1 cm. FTIR (Bruker Vector 22) having deuterated triglycine sulfate (DTGS) detector was used to record the spectra in the region of $400\text{--}4000$ cm^{-1} using KBr disk under 64 scans with a spectral resolution of 0.1 cm^{-1} .

To determine the particle size and zeta potential Nano-ZSP (Malvern Instruments) (zeta sizer) was used. Atomic Force Microscope (Agilent 5500) in tapping mode was used to record topographical images of PPG-AuNPs. For the preparation of analysis sample, one drop of the sample was placed on a silicon wafer that was freeze-dried for 24 hr. Triangular nitride silicon cantilever (Veeco, model MLCT-AUHW) was used for the analysis of the sample under a constant spring value of 0.1 Nm^{-1} .

2.2. Preparation of PPG-AuNPs

Solutions of $\text{HAuCl}_4 \cdot 3\text{H}_2\text{O}$ (0.25 mM) and NaBH_4 (5.0 mM) were prepared in deionized water and the solution of PPG-4000 (0.1 mM) was prepared in methanol. The ratio of the prepared solutions for preparation of PPG-AuNPs was kept 1:15:0.1 (PPG: $\text{HAuCl}_4 \cdot 3\text{H}_2\text{O}$: NaBH_4). PPG solution was stirred continuously while adding $\text{HAuCl}_4 \cdot 3\text{H}_2\text{O}$ solution to the flask followed by extra stirring for an hour. Afterward, the solution of NaBH_4 was added dropwise and the mixture was stirred for another 20 min. The appearance of the wine red color indicates the formation of

1
2
3 PPG-AuNPs. The effect of different external parameters which include temperature, pH, and
4 presence of electrolytic on the stability PPG-AuNPs evaluated.

8 **2.3. Application of PPG-AuNPs as a Colorimetric Sensor**

10 The aqueous solutions of CPH and other drugs having 0.1 mM concentration were prepared and
11 mixed with the solution of PPG-AuNPs to evaluate its effect on the visual change in color followed
12 by evaluation of variation in the SPR band by UV-visible spectroscopy.

15 **2.3.1. Analysis in Tap Water**

17 Tap water was taken from the University of Karachi. The above procedure was followed except
18 for the preparation of CPH solution which was prepared in tap water. The solution of PPG-AuNPs
19 and CPH were mixed, and the alteration in UV-spectra was observed.

22 **2.3.2. Analysis in Human Blood Plasma and Urine**

24 A blood sample was taken from the healthy human by using a venous puncture procedure with his
25 consent from the Center for Bioequivalence Studies and Clinical Research (CBSCR), International
26 Center for Chemical and Biological Sciences (ICCBS). Plasma was extracted from the blood by
27 centrifuging it at 4000 revolutions/min (rpm) for 5 min at 25 °C.

31 For the analysis, one control and other experimental solution of plasma were prepared. The control
32 contains 2.0 mL of plasma, 0.1 mL of PPG-AuNPs, and ~~makeup with~~ 10.0 mL distilled water.
33 In the experimental solution, 0.1 mM CPH was added, and afterward, UV-visible spectra of both
34 solutions were recorded. The same procedure was followed for the urine sample.

38 **2.4. Analysis in Pharmaceutical Drug**

40 For the analysis of CPH content in the pharmaceutical formulation, commercial drug samples were
41 purchased from the local market and a calculated amount was added into 2.0 mL of PPG-AuNPs.
42 After 5 min, the solutions were analyzed by UV-visible spectroscopy. Similar process was
43 employed to other commercial samples and concentrations were recorded using single point
44 analysis.

51 **3. Results and Discussion**

52 **3.1. Synthesis and Characterization of PPG-AuNPs**

54 PPG is a polyether with a pendant methyl group on the repeat unit. The ether groups in the polymer
55 chain induce dipole and polarity in the polymer which makes it a potential candidate as a stabilizing
56

agent for metal nanoparticles. A typical SPR band of AuNPs exists in the absorption range of 500-550 nm. The stoichiometric equivalence of groups responsible for stabilization on the stabilizing agent and AuNPs is important for optimum stabilization. In this case a ratio of 1:15 for solutions of PPG and $\text{HAuCl}_4 \cdot 3\text{H}_2\text{O}$ (v/v) found to be optimum, Figure 1A. The absorption decreased while going away from this ratio in either direction.²² The formation of AuNPs at the optimized ratios is further confirmed by AFM imaging, Figure 1B. The average size and zeta potential value of these prepared AuNPs calculated using DLS was found to be 104.6 nm and -8.0 mV respectively, Figure 5A (I,II,III).

Figure 1. (A) UV-visible spectra of PPG-AuNPs at the optimized ratio of PPG and $\text{HAuCl}_4 \cdot 3\text{H}_2\text{O}$ (1:15); (B) Three dimensional AFM image of PPG-AuNPs.

Next important aspect to evaluate of any prepared AuNPs is their stability when exposed to different experimental variables. Temperature treatment at 100 °C increased the absorption which is an indication of enhanced stability after thermal treatment, Figure 2A. Furthermore, PPG-AuNPs remain stable for more than a month at 4 °C. Another aspect to be evaluated is the stability of NPs in presence of electrolytes. Sodium chloride in a concentration range of 0.01 M to 5 M in PPG-AuNPs resulted in a decrease in the intensity of SPR band which may be attributed to aggregation of metal ions in presence of free chloride (Cl^-) ions, Figure 2B.²² The absorption band is not affected to a large extent while using NaCl concentration below 0.5 M. NaCl concentrations beyond this resulted in a significant drop in the intensity of the absorption band. On the same lines, the stability of PPG-AuNPs evaluated as a function of the pH. The maximum intensity of the SPR band found at pH 6. Changing pH in either direction leads to a decrease in the SPR band

intensity, however the selected pH of NPs for further studies is 5 as it is the original pH of prepared NPs, Figure 2C.²² Hence, PPG-AuNPs are found to be stable to different possible external parameters during its applications.

Figure 2. Stability of PPG-AuNPs through UV-visible spectroscopy (A) before and after incubation at 100 °C; (B) after adding different concentrations of NaCl; (C) at different pH.

3.2. PPG-AuNPs and Drug Interaction

Interaction of drugs of different nature with PPG-AuNPs evaluated by mixing equal volume of the solution of a drug having the same concentration as the AuNPs. The addition of fifteen tested drugs did not bring any physical change in color and the UV absorption band of PPG-AuNPs. However, the addition of ceftriaxone resulted in a color change from wine red to dark blue, and the SPR band shifts from 532 nm to 572 nm.²² Furthermore, the addition of CPH resulted in an abrupt disappearance of color and quenching of the SPR band of PPG-AuNPs, Figure 3. The abrupt disappearance of color and quenching of the SPR band is attributed to the aggregation of AuNPs. Structures of the drugs used in this study are shown in Figure 4.

Figure 3. UV-visible spectra of PPG-AuNPs before and after addition of different drugs, Inset show the color change after addition of CEF and CPH.

Figure 4. Structure of CPH and other tested drugs in this study

Moreover, AFM, zeta potential, and zeta sizer were used to evaluate the changes in PPG-AuNPs after the addition of CPH. A clear increase in size that is leading to aggregation of PPG-AuNPs is visible after the addition of CPH, Figure 5A-I, and Figure 5B-I. The mean average size and stability of particles were calculated by using zeta sizer and zeta potential, respectively. The average size of PPG-AuNPs was found to be 104.6 nm with a PDI of 0.395 that increased to 180.1 nm with a PDI of 0.201 after the addition of CPH, Figure 5A-II, and Figure 5B-II. Another important aspect is the net charge on the NPs that keeps them away from each other to avoid their aggregation. Net charge away from zero in either direction induce stability in the NPs. The net charge on the surface of PPG-AuNPs was found to be -8.07 mV which is an indication of reasonable stability of PPG-AuNPs. The net surface charge approaches zero after the addition of CPH which strongly suggests the aggregation of PPG-AuNPs, Figure 5A-III, and Figure 5B-III.

Figure 5. Evaluation of PPG-AuNPs after addition of CPH by (I) Atomic force micrographs (AFMs), (II) Average size, (III) the zeta potential of (A) PPG-AuNPs; (B) PPG-AuNPs/CPH

FTIR analysis of PPG, PPG-AuNPs, CPH, and PPG-AuNPs/CPH is carried out to determine the forces responsible for the stabilization of AuNPs and the interaction of the drug with PPG-AuNPs. The peaks of PPG at 3440 cm^{-1} (attributed to hydroxyl group stretching) shifted to 3340 cm^{-1} in the PPG-AuNPs spectrum which indicates the capping of AuNPs.²² The appearance of a peak at 1680

cm⁻¹ confirms the formation of Au-O bond. The addition of CPH in PPG-AuNPs did not affect the representative IR peak of AuNPs at 1680 cm⁻¹ which indicates the absence of any direct interaction of CPH with AuNPs. Nonetheless, the intensity of the peak at 1150 cm⁻¹ for -C-O decreases after addition of CPH which strongly indicates the interaction of CPH with oxygen present in the PPG. Meanwhile, in spectra of PPG-AuNPs and PPG-AuNPs/CPH, a clear change in the peak behavior was observed ranging from 2800 cm⁻¹ to 3000 cm⁻¹ which indicates the change in stretching of CH₃ and CH₂, see Figure 6.

Figure 6. FTIR spectra of PPG, PPG-AuNPs, CPH, and PPG-AuNPs/CPH

The addition of CPH in the PPG-AuNPs resulted in an immediate disappearance of typical wine red color and quenching of the SPR band. As shown above, electrostatic interactions are responsible for stabilization of AuNPs by PPG. The addition of CPH in stabilized PPG-AuNPs induce other competitive forces between ether groups of PPG and functional groups present on

CPH. CPH contains functional groups having the ability to donate electrons such as (C=O, C–O, N–H) which may interact strongly with the oxygen in the PPG chain leading to aggregation of AuNPs, Scheme 1. Another reason ¹ for abrupt quenching may be the transfer of electrons from AuNPs to cyclohexa-1, 4-diene moiety in CPH which is not present in other tested drug.

Scheme 1. Schematic representation of AuNPs formation through steric stabilization by the PPG and drug recognition (CPH) of PPG-AuNPs/CPH

3.3. Quantitative Recognition of CPH through PPG-AuNPs based Sensor

By recording UV spectra of PPG-AuNPs/CPH at different concentrations of CPH, the analytical capacity of the developed method is evaluated, Figure 7A. All of the used concentrations resulted in quenching of the SPR band. A careful analysis of the quenched peaks after the addition of CPH revealed a direct correlation between drug concentration and the extent of quenching Figure 7B. The absorbance at 532 nm is inversely proportional to the amount of CPH added. As a next step, a calibration curve is constructed by plotting absorbance at 532 nm as a function of the amount of CPH, Figure 7C. A reasonable linear correlation is found in the concentration range of 0.025-120 mM with the regression constant (R^2) value of 0.9983. The addition of 0.01 mM CPH solution did not affect the SPR band of PPG-AuNPs. The limit of detection [calculated by formula $3.0 \times (\text{SD of Intercept/Slope})$] of the proposed sensor was found to be 10.98 mM.^{44, 45}

Figure 7. (A) UV-Vis spectra of PPG-AuNPs after addition of the variable amount of CPH; (B) magnified version of Fig. 7A in a range of 0.12 to 0.162 absorbance; (C) absorbance intensity as a function of the amount of CPH

Immediate disappearance of the typical wine red color from PPG-AuNPs by the addition of CPH indicated the presence of CPH which is further confirmed by the disappearance/ quenching of SPR band. The coupling stoichiometry between PPG-AuNPs and CPH as obtained by Job plot was found to be 1:1, Figure 8.

Figure 8. Job plot for the binding ratio of PPG-AuNPs and CPH

The efficiency of any developed method must be validated in presence of other interfering drugs and salts in the context of real sample analysis. The developed sensor for CPH in this study is found to be robust in presence of several other interfering drugs and salts. The presence of other drugs namely aspirin, ibuprofen, ampiciline acid, paracetamol, clindamycin, aminophylline, cephalixin, salbutamol, phenytoin, isoniazid, azithromycin, cefixime, cefuroxime, DL tryptophan,

theophylline, and salts namely potassium bromide, calcium carbonate, calcium bicarbonate, and sodium chloride having concentration equal to CPH did not have any pronounced effect on the efficiency of the developed colorimetric CPH sensor, Figure 9.

Figure 9. UV-Vis spectra showing the effect of interfering drugs on CPH detection by PPG-AuNPs.

CPH analysis has been reported using many instrumental techniques such as HPLC, ELISA, voltammetry, NMR, and GC-MS in different environmental and biological samples, Table 1. Although the limit of detection of the instrumental methods is less compared to the instant colorimetric sensor proposed in this study, these methods require sophisticated instrumentation, highly equipped lab setup, expensive reagents, and trained operators. On the contrary, the proposed method in this study is facile, instantaneous, without any requirement of sample preparation, produce on spot results by visual demonstration. Furthermore, the obtained limit of detection is in a fairly low range in the context of practical applications. The evaluation of the efficiency of different interferences (drugs and salts) is an added advantage for such colorimetric sensors in the context of real sample analysis in minimum time without any high-tech instrumentation.

Table 1. Comparison of the analytical methods for the detection of CPH.

Method / Materials	Linear Range	*LoD	**LoD (mM)	Sample	Remarks	Ref.
Zero-Crossing Derivative Spectrophotometry	2.0-56.0 $\mu\text{g mL}^{-1}$	0.16 $\mu\text{g mL}^{-1}$	0.00045	saline physiological serum and Physiological serum-containing glucose	Conditions optimization, no interference study, complex statistical calculations required, no study in biological samples	¹³
Atomic Absorption Spectrometric Determination	5-70 $\mu\text{g mL}^{-1}$	6.69 $\mu\text{g mL}^{-1}$	0.01914	Drug samples	Lengthy conditions optimization, cost intensive, required heavy instrumentation, no interference study, no study in environmental and biological samples, trained operators required	⁴⁶
Electrochemical study of the degradation product	of 10^{-7} – 10^{-6} mol/L, with	0.5×10^{-7} mol/L.	0.00005	CPH solution	Instrument based method, selective electrodes required, limited accuracy, no interference study, trained operators required, no study in environmental and biological samples	⁷
Fluorescent supramolecular tweezers	0.1-5 μM	1 μM	0.001	Mixture of Drugs	Complicated protocol, proper laboratory setup required, expensive instruments required, condition optimization required, no study in environmental and biological samples	⁴⁷
Capillary Zone Electrophoresis	93.8–6255.6 mg/mL	5.0 $\mu\text{g mL}^{-1}$	0.01431	Mixture of Drugs	Tedious sample preparation, costly internal standards, long analysis time, optimization issues, proper laboratory setup and trained operators required	⁸
Spectrofluorimetric method	0.1–5.0 $\mu\text{g/mL}$.	$1.09 \times 10^{-2} \pm 3.64 \times 10^{-3}$ $\mu\text{g/mL}$	0.000028	Commercial Formulations	Optimization of Conditions, sophisticated instrumentation, trained operators required, cost intensive method	⁶

Fluorosurfactant-capped gold nanoparticles	2.0–10.0 mg mL ⁻¹	0.8 µg mL ⁻¹	0.00228	Pharmaceutical formulations	No study in environmental and biological samples, usage of costly reagents, no interference studies presence of similar nature drugs and electrolytes	⁴⁸
High-performance liquid chromatographic method	0.2 to 30.0 µg/ml	0.2 µg/ml	0.000572	Human Plasma samples	Exclusive instrumentation, trained operators, long analysis time, lengthy sample preparation, optimization issues, long analysis time, expensive standards required	⁴
Spectrophotometric/PPG-AuNPs Method	0.025-120 mM	10.98 mM	10.980	Pharmaceutical formulations, water, blood plasma, Urine, Serum	Simple procedure, reasonable sensitivity, high selectivity, short analysis time, basic instrumentation, economical, on-spot visual indication, no pre-treatment of the sample	This method

*LoD in the units as given in the reference

**LoD unit converted to mM

Up till now, all the above studies were carried out in DI water, the authenticity of interference study and validity of PPG-AuNPs based proposed assay for CPH in context of real samples evaluated by employing the optimized protocols for tap water, urine, serum, and blood plasma samples. Immediate disappearance of wine red color of PPG-AuNPs and quenching of the typical UV peak confirms the applicability of the proposed CPH assay for all real samples in presence of natural interferents, Figure 10. The extent of quenching in the real samples follows the same calibration curve and hence the assay is proved to be quantitative for the real samples in tap water, urine, serum, and blood plasma.

Figure 10. Effect of addition of CPH on absorption band of PPG-AuNPs (A) tap water; (B) blood plasma; (C) Urine; (D) Serum.

As a next step, the developed CPH assay is employed for quantification and quality control of commercial drugs containing CPH. An appropriate amount of the commercial drug as per claims of the manufacturer added in the PPG-AuNPs. The immediate disappearance of the wine-red color of the PPG-AuNPs gave the first indication of the presence of CPH. The next step is the evaluation of the quantitation capability of the proposed colorimetric sensor. The extent of

quenching in the UV absorption band, evaluated by the absorbance after addition of the drug, is compared with the calibration curve, Figure 7C. The amount of CPH as obtained by the calibration curve is fairly close to the manufacturer claim, Table 2.

Table 2. Quantitative analysis of CPH containing commercial samples with regard to claimed content of CPH

Sample	Total amount of tablet (mg)	Amount of cephhradine (mg)	Mean Abs. values	Calc. Conc. of CPH (mM)	Calc. amount of CPH (mg)	Calc. amount of CPH in tablet (mg)	Manufacturer claim CPH (mg)
Standard	-	0.4506	0.3436	0.5005	-	-	-
Sample 1	590.2120	0.4506	0.3230	0.4704	0.3287	430.5430	500
Sample 2	295.8713	0.4506	0.3030	0.4413	0.3083	215.8297	250

The proposed colorimetric assay for CPH in this study is robust and applicable to samples containing different interfering species and for quick quality control in the production facilities. The advantages of the proposed colorimetric assay for CPH include fast, facile, on-spot analysis of drug contamination in environmental and biological samples without any involvement of high-tech instrumentation.

4. Conclusion

The uniquely developed colorimetric sensing probe consisting of PPG-AuNPs proved to be a rapid and sophisticated one-pot quantitative assay for CPH. Characterization of PPG-AuNPs, CPH, and PPG-AuNPs/CPH was done through zeta sizer, AFM, FTIR, and UV-visible spectroscopy. The synthesized PPG-AuNPs were found to be resistant to various external variables such as temperature, pH, and presence of electrolytes. Typical wine red color of AuNPs immediately disappears by addition of Sr^{2+} and SPR band of AuNPs. The proposed aggregation sensor is efficient in a dynamic range of 0.025-120 mM having limit of detection of 10.7 nM. The presence of other drugs and ions did not have any pronounced effect on sensing ability for CPH. The proposed sensor worked efficiently for environmental, biological, and pharmaceutical samples. Excellent selectivity, simple procedure of preparation and application, a reasonable dynamic range and limit of detection demonstrate potential of PPG-AuNPs based CPH sensor for practical applications without any involvement of high-tech instrumentation requiring minimal laboratory

set-up. Furthermore, the proposed method can be employed as a quick screening of pharmaceutical formulations in context of quantification of CPH on production facilities.

Data Availability: Data is available at https://datadryad.org/stash/share/TF8Xb4NqiQ8CeD9PbSN2C0jq5zk_t5Qe9758r4SMj_A

Conflict of Interest: The authors declare no conflict of interest.

Funding: There is no external funding to report for this project.

Author Contributions:

Daim Asif Raja performed experiments and wrote the initial version of the manuscript,

Fazeelah Munir performed experiments and wrote the initial version of the manuscript,

Daim Asif Raja and Fazeelah Munir have equal contribution in this study and both should be considered as first authors.

Muhammad Raza Shah provided facilities for the performed work along with intellectual input,

Muhammad Iqbal Bhangar provided intellectual input and help writing manuscript,

Muhammad Imran Malik conceived the idea and finalized manuscript

5. References

1. E. Martínez-Carballo, C. González-Barreiro, S. Scharf and O. Gans, *Environ. Pollut*, 2007, **148**, 570-579.
2. J. L. Martinez, *Environ. Pollut*, 2009, **157**, 2893-2902.
3. K. Florey, in *Analytical profiles of drug substances*, Elsevier, 1976, vol. 5, pp. 21-59.
4. V. M. Johnson, J. P. Allanson and R. C. Causon, *J. Chromatogr. B: Biomed. Sci. Appl.*, 2000, **740**, 71-80.
5. M. Abdel-Hamid, M. Mahrous, H. Daabees and Y. Beltagy, *J. Clin. Pharm. Ther.*, 1992, **17**, 91-95.
6. J. Shah, M. R. Jan, S. Shah and S. N. A. Shah, *J. Anal. Chem.*, 2014, **69**, 638-645.
7. Q. Jiang, Y. Ying, J. Wang, Z. Ye and Y. Li, 2005.
8. Y. Shen, H. Liu, S. Rong, Y. Li and C. Hu, *Anal. Lett.*, 2006, **39**, 569-578.
9. M. Mascini, S. Gaggiotti, F. Della Pelle, J. Wang, J. M. Pingarrón and D. Compagnone, *Biosens. Bioelectron*, 2019, **123**, 124-130.
10. A. Masri, A. Anwar, D. Ahmed, R. B. Siddiqui, M. Raza Shah and N. A. Khan, *J. Antibiot*, 2018, **7**, 100.
11. S. S. Memon, A. Nafady, A. R. Solangi, A. M. Al-Enizi, M. R. Shah, S. T. Sherazi, S. Memon, M. Arain, M. I. Abro and M. I. Khattak, *Sens. Actuators, B*, 2018, **259**, 1006-1012.
12. A. Minhaz, M. Ishaq, I. Ahmad, F. Ahmed and M. R. Shah, *Sens. Lett*, 2016, **14**, 310-318.
13. J. Murill, J. Lemus and L. Garci, *Ana. Lett.*, 1994, **27**, 1875-1892.
14. S. H. Qaddare and A. Salimi, *Biosens. Bioelectron*, 2017, **89**, 773-780.

15. A.-C. Burduşel, O. Gherasim, A. M. Grumezescu, L. Mogoantă, A. Fikai and E. Andronescu, *Nanomaterials*, 2018, **8**, 681.
16. N. Elahi, M. Kamali and M. H. Baghersad, *Talanta*, 2018, **184**, 537-556.
17. P. N. Njoki, I.-I. S. Lim, D. Mott, H.-Y. Park, B. Khan, S. Mishra, R. Sujakumar, J. Luo and C.-J. Zhong, *J. Phys. Chem. C*, 2007, **111**, 14664-14669.
18. R. Herizchi, E. Abbasi, M. Milani and A. Akbarzadeh, *Artif. Cells Nanomed. Biotechnol.*, 2016, **44**, 596-602.
19. D. P. Stankus, S. E. Lohse, J. E. Hutchison and J. A. Nason, *Environ. Sci. Technol.*, 2011, **45**, 3238-3244.
20. A. Franconetti, J. M. Carnerero, R. Prado-Gotor, F. Cabrera-Escribano and C. Jaime, *Carbohydr. Polym.*, 2019, **207**, 806-814.
21. M. N. Nichick, S. V. Voitekhovich, A. Shavel, A. I. Lesnikovich and O. A. Ivashkevich, *Polyhedron*, 2009, **28**, 3138-3142.
22. D. A. Raja, S. G. Musharraf, M. R. Shah, A. Jabbar, M. I. Bhangar and M. I. Malik, *J. Ind. Eng. Chem.*, 2020.
23. Y. Xu, F. Y. Kutsanedzie, M. Hassan, J. Zhu, W. Ahmad, H. Li and Q. Chen, *Food Chem.*, 2020, **315**, 126300.
24. S. Laksee, K. Sansanaphongpricha, S. Puthong, N. Sangphech, T. Palaga and N. Muangsin, *Int. J. Biol. Macromol.*, 2020, **162**, 561-577.
25. D. Cabuzu, A. Cirja, R. Puiu and A. Mihai Grumezescu, *Curr. Top. Med. Chem.*, 2015, **15**, 1605-1613.
26. K. Sztandera, M. Gorzkiewicz and B. Klajnert-Maculewicz, *Mol. Pharmaceutics*, 2018, **16**, 1-23.
27. A. J. Mieszawska, W. J. Mulder, Z. A. Fayad and D. P. Cormode, *Mol. Pharmaceutics*, 2013, **10**, 831-847.
28. Y. Liu, H. Miyoshi and M. Nakamura, *Int. J. Cancer*, 2007, **120**, 2527-2537.
29. S. Rahim, A. Rauf, S. Rauf, M. R. Shah and M. I. Malik, *RSC Adv.*, 2018, **8**, 35776-35786.
30. L. E. Silva-De Hoyos, V. Sanchez-Mendieta, M. A. Camacho-Lopez, J. Trujillo-Reyes and A. R. Vilchis-Nestor, *Arabian J. Chem.*, 2020, **13**, 1975-1985.
31. L. Bach, M. Thi, N. Son, Q. Bui, H.-T. Nhac-Vu and P. Ai-Le, *J. Electroanal. Chem.*, 2019, **848**, 113359.
32. A. Scroccarello, F. Della Pelle, L. Neri, P. Pittia and D. Compagnone, *Food Res. Int.*, 2019, **119**, 359-368.
33. S. Chah, M. R. Hammond and R. N. Zare, *Chem. Biol.*, 2005, **12**, 323-328.
34. C. Sönnichsen, B. M. Reinhard, J. Liphardt and A. P. Alivisatos, *Nat. Biotechnol.*, 2005, **23**, 741-745.
35. N. L. Rosi, D. A. Giljohann, C. S. Thaxton, A. K. Lytton-Jean, M. S. Han and C. A. Mirkin, *Science*, 2006, **312**, 1027-1030.
36. N. ul Ain, Z. Aslam, M. Yousuf, W. A. Waseem, S. Bano, I. Anis, F. Ahmed, S. Faizi, M. I. Malik and M. R. Shah, *New J. Chem.*, 2019, **43**, 1972-1979.
37. F. Ikram, A. Qayoom and M. R. Shah, *Sens. Actuators, B*, 2018, **257**, 897-905.
38. S. Rahim, S. A. Ali, F. Ahmed, M. Imran, M. R. Shah and M. I. Malik, *J. Nanopart. Res.*, 2017, **19**, 259.
39. S. Rahim, S. Khalid, M. I. Bhangar, M. R. Shah and M. I. Malik, *Sens. Actuators, B*, 2018, **259**, 878-887.
40. Y. Xue, B. Dong, X. Liu, F. Wang, J. Yang and D. Liu, *Sci. China Chem.*, 2019, **62**, 280-286.
41. H. Shinzawa, T. Uchimarui, J. Mizukado and S. G. Kazarian, *Vib. Spectrosc.*, 2017, **88**, 49-55.
42. M. Rajan, R. A. Praphakar, D. Govindaraj, P. Arulselvan and S. S. Kumar, *Mater. Today Chem.*, 2017, **6**, 26-33.

- 1
- 2
- 3
- 4 43. M. Saranya, L. Srinivasan, G. R. MR, T. Gomathi, P. Sudha and A. Sukumaran, *Int. J. Biol. Macromol.*, 2017, **104**, 1436-1448.
- 5
- 6 44. S. Rahim, A. M. Bhayo, M. R. Shah and M. I. Malik, *Microchem. J.*, 2019, **149**, 104048.
- 7 45. H. Niu, Y. Yang and H. Zhang, *Biosens. Bioelectron.*, 2015, **74**, 440-446.
- 8 46. S. M. Al-Ghannam, *J. Food Drug Anal.*, 2008, **16**, 19.
- 9 47. B. Khan, A. Minhaz, I. Ali, S. Nadeem, S. Yousuf, M. Ishaq and M. R. Shah, *Tetrahedron Lett.*, 2015, **56**, 581-585.
- 10
- 11 48. C. Lu, N. Zhang, J. Li and Q. Li, *Talanta*, 2010, **81**, 698-702.
- 12
- 13
- 14
- 15
- 16
- 17
- 18
- 19
- 20
- 21
- 22
- 23
- 24
- 25
- 26
- 27
- 28
- 29
- 30
- 31
- 32
- 33
- 34
- 35
- 36
- 37
- 38
- 39
- 40
- 41
- 42
- 43
- 44
- 45
- 46
- 47
- 48
- 49
- 50
- 51
- 52
- 53
- 54
- 55
- 56
- 57
- 58
- 59
- 60

Appendix B

Response to Reviewer's Comments

Reviewer: 1

All corrections regarding spelling and grammar as suggested by reviewer 1 in the annotated PDF are adopted in the current version.

Before or after the drop placement? (P5, L39)

Air-dried after placement of sample. Added in the manuscript too

How can NP concentration be expressed and compared with the drug concentration? (P8, L42)

Equal volumes of the mentioned concentrations were used for the sake of comparison.

Explain abbreviation d.nm used in the figure

(d.nm) is Diameter in nanometer of nanoparticles. Added in the manuscript too

I do not understand this figure. Spectra are similar for CPH and other drugs. This should be better explained (P15 L33)

This figure presents the merit of the developed method. It represents the efficiency of the developed sensor in the presence of other drugs and salts in the solution which can be present in the real samples. Added couple of sentences in the manuscript to clarify more as suggested by the reviewer.

“The efficiency of any developed method must be validated in the presence of other interfering drugs and salts in the context of real sample analysis. The developed sensor for CPH in this study was found to be robust in the presence of several other interfering drugs and salts. The presence of other drugs namely aspirin, ibuprofen, ampiciline acid, paracetamol, clindamycin, aminophylline, cephalexin, salbutamol, phenytoin, isoniazid, azithromycin, cefixime, cefuroxime, DL tryptophan, theophylline, and salts namely potassium bromide, calcium carbonate, calcium bicarbonate, and sodium chloride having concentration equal to CPH did not have any pronounced effect on the efficiency of the developed colorimetric CPH sensor. Hence, the efficiency of the proposed quenching sensor for CPH was not affected by presence of other competing drugs and salts which is an advantage in context of real sample analysis. “

It should be good to use a validated method for comparison, not just manufacturer claim (P19 L7)

The results of quantification of CPH in commercial drugs are now validated by HPLC and UV-Vis spectroscopy and results are compared in the Table 2.

141 use reasonable number of valid digits in LOD and LOQ as well (P19 L45)

Thanks for pointing out. The number of significant figures is now reduced to valid digits as suggested by the reviewer.

Reviewer: 2

1. In Fig.5 (III), please specify the solution condition (concentration, pH, etc.) due to zeta potential is sensitive to the solution condition

The conditions are now added at the mentioned point as suggested by the reviewer.

PPG (0.1 mM): HAuCl₄.3H₂O (0.25 mM): NaBH₄ (5.0 mM) = 1:15:0.1 (v/v); pH (~5)

2. In Fig.7C and Fig. 8, for more clearly indicating the ordinate, the name of ordinate should be “absorbance at 532 nm” instead of just “absorbance”.

Modified as suggested by the reviewer

3. In order to prove the accuracy of proposed method, please include the recovery and relative standard deviation (RSD) in Table 2.

Table 2 is now modified as suggested by the reviewer 1 and 2.

4. As the authors said the PPG increase the hydrophobic character compared to PEG, the PPG-AuNPs may have the lower stability in aqueous solution, what about the stability of PPG-AuNPs solution ?

The presence of methyl group on the repeat unit increase the hydrophobicity of PPG as compared to PEG. However, PPG is readily miscible with methanol and water, as the value of zeta potential is -8.0 mV which shows the good stability of PPG-AuNPs. PPG-AuNPs remain stable for more than one month at 4 °C.